# Campus healthcare workers' knowledge and views on injectable pre-exposure prophylaxis prior to rollout in South Africa: A qualitative study

Nomvuselelo Nomzamo Mbatha[1]*, Nomakhosi Mpofana[2], Dumile Gumede[1]

1 Faculty of Health Sciences, Durban University of Technology, Durban, South Africa, 2 Department of Somatology, Faculty of Health Sciences, Durban University of Technology, Durban, South Africa

* 21518444@dut4life.ac.za

## Abstract

Long-acting injectable pre-exposure prophylaxis (PrEP) offers a promising alternative to daily oral PrEP by addressing adherence challenges. However, successful implementation depends on the readiness and perceptions of primary healthcare (PHC) workers who deliver these services, particularly in campus health clinics serving high-risk youth populations. The objective of this study was to explore PHC workers' knowledge and views on long-acting injectable HIV PrEP in campus health clinics before its rollout in South Africa. We conducted a qualitative, exploratory, cross-sectional study using semi-structured interviews with a purposive sample of 18 PHC workers from six campus health clinics at a public university in KwaZulu-Natal, South Africa. Data were collected between August and September 2025, and were analyzed thematically using Babchuk's 10-step process. PHC workers demonstrated limited and variable awareness of injectable PrEP across professional cadres. While participants anticipated improvements in HIV prevention through injectable PrEP regarding adherence and reducing stigma, they expressed confusion about the clinical protocol surrounding injectable PrEP rollout; however, they were hopeful, dependent on the demonstrated success of injectable PrEP. PHC workers currently lack the preparedness required for injectable PrEP delivery. To enable effective rollout, the government must prioritize comprehensive training on clinical and operational protocols.

## Background

Globally, approximately 4,200 young women aged 15–24 years acquire HIV each week, with the highest incidence occurring in sub-Saharan Africa [1]. South Africa bears the world's largest HIV burden, and young people remain disproportionately affected despite extensive national prevention programmes [1,2]. KwaZulu-Natal

**Data availability statement:** The datasets generated and/or analyzed during the current study are not publicly available due to confidentiality agreements, but data requests can be made to the Research Ethics Committee at irec@ dut.ac.za. This study is qualitative in nature. The minimal dataset consists of anonymized interview excerpts, which are presented directly within the paper to support the reported findings. Full transcripts cannot be made publicly available as they contain information that could compromise participant confidentiality, consistent with the ethical approval granted by the Institutional Research Ethics Committee (IREC Reference: IREC 113/25). The IREC contact (irec@dut.ac.za) is an independent, institutional, non-author's point of contact who can field any data inquiries from fellow researchers.

**Funding:** This research was supported by the Department of Higher Education and Training Nurturing Emerging Scholars Program (DHET NESP), South Africa. The funder had no role in study design, data collection, analysis, interpretation, or manuscript preparation.

**Competing interests:** The authors have declared that no competing interests exist.

consistently records the highest HIV prevalence nationally, underscoring the urgent need for differentiated, youth-centered prevention approaches [3].

Since 2016, daily oral pre-exposure prophylaxis (PrEP) has been available in South Africa [4]. However, its effectiveness is often undermined by adherence challenges among young people, who struggle with daily pill-taking due to pill fatigue, stigma, privacy concerns, and lifestyle patterns [5,6]. These persistent adherence barriers have intensified calls for HIV prevention options that reduce dosing frequency and minimize user burden.

Long-acting injectable PrEP (cabotegravir), administered every two months, has demonstrated superior efficacy compared to daily oral PrEP in multiple clinical trials [7,8]. In 2022, the WHO recommended its use, and South Africa subsequently approved a phased rollout prioritizing youth-friendly settings [9,10]. While this represents a major advancement in HIV prevention, its successful adoption will depend heavily on the readiness and perceptions of healthcare workers who are central to implementation.

Campus health clinics are strategic access points for HIV prevention among university students, a population characterized by high rates of concurrent partnerships, inconsistent condom use, experiences of gender-based violence, and limited access to tailored sexual health services [11,12]. Unlike general primary healthcare settings, campus clinics serve a concentrated, mobile, and socially interconnected population of young adults, making them uniquely positioned yet operationally distinct environments for delivering injectable PrEP. All 26 public universities in South Africa maintain such clinics staffed by PHC workers who already provide HIV testing, counselling, contraception, STI services, and increasingly, oral PrEP [13]. As injectable PrEP becomes available, these clinics are expected to assume a central role in youth-focused implementation efforts.

Successful uptake of new health technologies relies significantly on provider readiness, encompassing knowledge, attitudes, perceived capacity, and systemic support [14,15]. Healthcare workers act as gatekeepers in HIV prevention, with their preparedness strongly influencing client uptake, service quality, and long-term sustainability [16,17]. However, existing studies on injectable PrEP have largely examined the perspectives of potential users or healthcare workers in general PHC settings, with limited attention to campus-based providers, whose operational realities and student populations differ markedly [18,19].

This gap is significant because campus health clinics face unique constraints, including fluctuating student demand driven by academic calendars, heightened privacy concerns within close-knit campus communities, and resource limitations specific to educational institutions [20]. Understanding PHC workers' knowledge, attitudes, and perceived barriers and facilitators within this context is essential for designing effective implementation strategies.

This study explored PHC workers' knowledge, attitudes, and beliefs regarding injectable PrEP in campus health clinics. This study aims to generate evidence that directly informs implementation planning for South Africa's upcoming rollout of injectable PrEP.

## Methods

### Study design

We conducted a qualitative, exploratory, cross-sectional study underpinned by an interpretivist paradigm, which emphasizes understanding individuals' subjective meanings and experiences within their social contexts [21]. This article is partially based on the first author's master's project.

### Study setting

The study was conducted at six campus health clinics of a large public university in KwaZulu-Natal, South Africa. These clinics provide comprehensive primary healthcare services to predominantly young adult students (aged 18–30 years), including HIV prevention, testing, and treatment; family planning; sexually transmitted infection (STI) screening and management; and general primary care. Each clinic is staffed by multidisciplinary teams comprising PHC nurses, part-time medical doctors, HIV counsellors, and health promoters. At the time of this study, all six clinics offered oral PrEP services but had not yet introduced injectable PrEP.

### Participants and sampling

We employed purposive sampling to recruit PHC workers with relevant knowledge and experience in HIV prevention service delivery [22]. Eligible participants were: (a) employed at one of the six campus health clinics; (b) aged ≥18 years; (c) directly involved in HIV prevention services, including oral PrEP delivery; and (d) willing to provide informed consent. We excluded healthcare workers not involved in HIV services.

Recruitment was facilitated through clinic heads who introduced the study to eligible staff members. An informational flyer outlining study purpose, eligibility criteria, confidentiality assurances, and researcher contact details was distributed in staff common areas. Interested individuals contacted the researcher or expressed interest through clinic heads, after which they received detailed verbal and written study information.

Sample size was guided by the principle of information power [23], with sampling continuing until data saturation, the point at which no new themes or insights emerged, was achieved. Saturation was reached after 16 interviews; two additional interviews were conducted to confirm that no new patterns emerged, resulting in a final sample of 18 participants.

### Data collection

Semi-structured, face-to-face interviews were conducted between the 11th of August 2025 and 09th of September 2025 by the first author (NNM), a female Master's student trained in qualitative research methods. Interviews lasted 45–60 minutes and were conducted in private clinic consultation rooms to ensure confidentiality. All interviews were conducted one-on-one with no other persons present besides the participant and interviewer. No repeat interviews were conducted. All healthcare workers approached agreed to participate in the study; there were no refusals or non-responses. Participants chose to be interviewed in English. The interview guide (S1 Appendix) was developed based on Rogers' Diffusion of Innovations constructs [24] and piloted with three PHC workers from non-study clinics to refine question clarity and flow. Pilot data were not included in the analysis.

All interviews were audio-recorded with written consent and supplemented by field notes capturing non-verbal cues, contextual observations, and interviewer reflections. Following each interview, audio recordings were stored on password-protected devices, and the researcher conducted debriefing sessions with supervisors (DG, NM) to reflect on the interview process, assess data quality, and maintain reflexivity.

## Data analysis

Audio recordings were transcribed verbatim by the first author, with transcripts verified for accuracy by the co-authors. Data were managed and analyzed using NVivo 15 software (QSR International). We employed Babchuk's 10-step thematic analysis process [25], which combines systematic rigor with interpretive flexibility.

This combined deductive and inductive approach ensured theoretical grounding while remaining open to unexpected findings. A detailed codebook was developed iteratively, with codes defined, refined, and organized hierarchically. Preliminary themes were not shared with participants for feedback.

## Trustworthiness

Trustworthiness was established using Lincoln and Guba's criteria [26]. Credibility was ensured through prolonged engagement with data through multiple analysis cycles, triangulation across participant roles, supervisor debriefing sessions, and use of verbatim quotations. Dependability was maintained through a comprehensive audit trail of raw transcripts, coding frameworks, field notes, analytic memos, and successive thematic iterations. Confirmability was achieved through reflexive journaling documenting researcher's positionality and potential biases, grounding interpretations in participant data rather than researcher assumptions. Transferability was supported by a thick description of the study context, setting, participants, and procedures, enabling readers to assess applicability to other contexts. Authenticity was ensured through fair representation of diverse perspectives, including minority views, and attention to power dynamics, ensuring marginalized voices were acknowledged.

## Ethical considerations

Ethical approval was obtained from the Institutional Research Ethics Committee (IREC) (Reference: IREC 113/25). Gatekeeper permission was secured from relevant university stakeholders. All participants provided written informed consent before participation in the study.

## Results

### Participant characteristics

The 18 participants comprised 10 PHC nurses (55.6%), 3 medical doctors (16.7%), and 5 HIV counsellors (27.8%), predominantly female (88.9%) and mid-to-late career (mean age 52.4 years, SD 13.8; mean clinical experience 16.6 years, SD 11.2). All participants were directly involved in HIV prevention services, with 83.3% having more than one year of oral PrEP experience and 88.9% having previous injectable medication training, predominantly family planning injections (72.2%). Detailed characteristics are presented in Table 1.

### Themes

Four major themes emerged from the analysis: (1) variations in knowledge across healthcare worker categories, (2) confusion about clinical and operational protocols surrounding injectable PrEP rollout, (3) anticipated improvements in HIV prevention through injectable PrEP, and (4) hopeful, dependent on demonstrated success. Throughout this section, findings are supported by participant verbatim quotes with only minor grammatical adjustments for clarity.

#### Theme 1: Variations in knowledge across healthcare worker categories

The findings indicate notable differences in awareness and understanding of injectable PrEP among various healthcare worker categories within campus health clinics. Among the ten nurses interviewed, seven reported having some knowledge of injectable PrEP, whereas three indicated limited or no awareness of it. The three doctors demonstrated a moderate level of awareness overall, with one displaying a clear understanding of the intervention and two acknowledging only limited

**Table 1. Demographic characteristics of study participants (n = 18).**

| Participant code | Age (years) | Sex | Occupation | Years of Experience |
|---|---|---|---|---|
| P1 | 70 | Female | PHC Nurse | 30 |
| P2 | 68 | Female | PHC Nurse | 27 |
| P3 | 39 | Female | PHC Nurse | 12 |
| P4 | 35 | Female | PHC Nurse | 03 |
| P5 | 27 | Female | PHC Nurse | 05 |
| P6 | 62 | Female | PHC Nurse | 30 |
| P7 | 46 | Female | PHC Nurse | 08 |
| P8 | 67 | Female | PHC Nurse | 28 |
| P9 | 57 | Female | PHC Nurse | 11 |
| P10 | 66 | Female | PHC Nurse | 35 |
| P11 | 50 | Female | Medical Doctor | 30 |
| P12 | 50 | Male | Medical Doctor | 23 |
| P13 | 38 | Female | Medical Doctor | 15 |
| P14 | 55 | Female | HIV Counsellor | 01 |
| P15 | 55 | Female | HIV Counsellor | 15 |
| P16 | 32 | Male | HIV Counsellor | 06 |
| P17 | 64 | Female | HIV Counsellor | 02 |
| P18 | 42 | Female | HIV Counsellor | 19 |

*Abbreviations: PHC = Primary healthcare; PrEP = Pre-exposure prophylaxis; SD = Standard deviation*

familiarity. Among the five HIV counsellors, four reported that they had heard about injectable PrEP, though their knowledge was largely surface-level, and one counsellor indicated minimal awareness. Some participants demonstrated greater knowledge with the injectable form of PrEP and highlighted its advantage over oral PrEP, as described in the excerpts below:

> *"Injectable PrEP is cabotegravir administered intramuscularly every two months. It provides sustained protection against HIV acquisition and has shown high efficacy in clinical trials. The main advantage over oral PrEP is that it eliminates the daily pill burden, which should theoretically improve adherence. However, I think we need more local data on its effectiveness in our population and how patients respond to it in our local setting."* [P12, Medical Doctor]

In contrast, other participants reported only a basic awareness of injectable PrEP and lacked understanding of its recommended dosing intervals and associated side effects, highlighting uncertainty in their confidence in future service delivery.

> *"I've heard about injectable PrEP, but most of what I know comes from online articles. We haven't really had formal training yet. I know it's supposed to prevent HIV, but I'm not clear on how often patients need to come back or what the side effects might be. It would help if we had proper sessions where someone explains everything in detail."* [P05, PHC Nurse]

Similarly, an HIV counsellor explained that they currently lack the information needed to guide clients in choosing between different PrEP modalities:

> *"I've heard colleagues talking about it, and I saw something about it on the news last year [2024]. From what I understand, it's an injection that protects against HIV for a few months. But honestly, I don't know much about who should get*

*it, whether it has serious side effects, or how we would counsel students about choosing between pills and injections. I'd need proper training before I could confidently discuss it with clients."* [P16, HIV Counsellor]

Taken together, although participants' depth of knowledge varied, there was a shared understanding that injectable PrEP represents a new development in HIV prevention, with information most often obtained through online news and peer networks. Most participants also emphasized the need for formal training and structured information on injectable PrEP to strengthen their confidence in supporting its implementation.

### Theme 2: Confusion about clinical and operational protocols surrounding injectable PrEP rollout

Although most participants recognized injectable PrEP as a new HIV prevention method, they expressed uncertainty about its clinical requirements and operational implementation within campus health clinics. Knowledge gaps included its effectiveness, the duration of protection, dosage schedules, and the specific eligibility criteria for potential users.

Some participants expressed uncertainty about the effectiveness and dosing requirements of injectable PrEP. Although aware that the intervention is intended to prevent HIV, they were unsure how well it works in comparison to existing oral PrEP and did not feel confident explaining its protective level to clients. Participants were also unclear about how frequently injections would be required and whether adherence demands would differ from daily oral PrEP. One participant said:

*"I'm not sure if it's already in South Africa or just something still being tested. And even if it is available, I don't know how long one injection lasts. Is it monthly? Every three months? And does it work as well as the tablets? These are the kinds of questions students will ask us, and right now, I wouldn't have good answers for them."* [P07, PHC Nurse]

Other participants also voiced uncertainty about the eligibility criteria for accessing injectable PrEP. While they understood that the intervention is intended for HIV prevention, many were unsure about the specific clinical guidelines that determine who can receive it.

*"Some patients ask about new prevention methods, but we don't have enough information to guide them properly. For example, I don't know if injectable PrEP is only for people at very high risk or if any student can request it. Can someone on chronic medication also take it? What about someone who's already on oral PrEP, can they switch? Without clear guidelines, it's hard to counsel effectively."* [P18, HIV Counsellor]

As a result, participants' limited understanding of eligibility guidelines created uncertainty and hesitation around how to effectively identify and support clients who could be suitable candidates for injectable PrEP.

### Theme 3: Anticipated improvements in HIV prevention through injectable PrEP

Participants highlighted the convenience and long-acting nature of injectable PrEP as key advantages that could improve client uptake and adherence. They perceived it as an innovation that might particularly benefit individuals who are concerned about maintaining daily pill-taking routines, stigma related to oral PrEP use or privacy issues. As one participant said:

*"We feel hopeful about injectable PrEP because it could really change the game for prevention. Students often forget to take their pills daily, or they stop because they don't want others to see them taking medication. With an injection every two months, they don't have to worry about pills at all. It's more discreet, and I think it would encourage more students to protect themselves."* [P15, HIV Counsellor]

This optimism was grounded in the perception that injectable PrEP offers individuals greater autonomy to choose between a pill or an injection, similar to the range of options available for contraceptive methods. Participants felt that such a choice could empower users, allowing them to select a prevention method that best fits their personal preferences and lifestyles. In the words of another participant:

*"This is a welcome addition to the HIV prevention methods. Anything that improves adherence is beneficial, and I believe injectable PrEP has that potential. We've seen with family planning injections that students prefer them over daily pills. I think the same could be true for HIV prevention methods. If we can offer both options, pills and injections, then clients can choose what works best for their lifestyle."* [P02, PHC Nurse]

In summary, participants perceived injectable PrEP as an innovation that could enhance overall prevention outcomes by supporting better adherence and enabling individuals to choose their preferred prevention modality.

**Theme 4: Hopeful, dependent on demonstrated success**

While participants recognised the potential benefits of introducing injectable PrEP, some expressed a reserved stance, noting that their confidence would depend on seeing how well the method performs in reality. These participants recognized the promise of injectable PrEP in strengthening HIV prevention efforts, while also acknowledging their doubts associated with the nature of the target population for injectable PrEP. One doctor captured this sentiment:

*"Some of us are excited about injectable PrEP, but others are still hesitant until they see actual results. It sounds good in theory, convenient, long-acting, and better adherence. But we need to see how it works in practice. Will there be side effects we haven't anticipated? I'm cautiously optimistic, but I think we need to approach this thoughtfully and monitor outcomes closely."* [P12, Medical Doctor]

Another participant shared similar views, emphasizing the need for demonstrated effectiveness before full commitment:

*"I think it's a good idea, but we have to see if it really works as promised. We've had new interventions before that looked great on paper but didn't perform well in our setting. So while I'm hopeful about injectable PrEP, I want to see evidence from our own clinics, from our own students, before I'm fully convinced. Show me the results, and then I'll be a strong advocate for it."* [P11, Medical Doctor]

Participants acknowledged the value of expanding HIV prevention options but highlighted that practical evidence from campus health contexts would be necessary before they could fully advocate for its use.

Overall, while participants were receptive to the innovation, full endorsement of injectable PrEP remained contingent on demonstrated success in practice, particularly within campus health settings where patterns of service utilization may differ from those observed in clinical trials.

## Discussion

This study explored primary healthcare workers' knowledge, attitudes, and beliefs about long-acting injectable HIV PrEP in campus health clinics at a public university in KwaZulu-Natal, South Africa. Through semi-structured interviews with 18 PHC workers, we sought to understand their knowledge and views regarding this emerging HIV prevention technology before its national rollout in South Africa.

Four key findings emerged from the analysis. First, PHC workers demonstrated notable variations in knowledge across professional categories, with awareness ranging from a comprehensive understanding among some doctors to surface-level familiarity among most counsellors and nurses. Second, significant clinical protocol confusion surrounded

the injectable PrEP rollout, with participants expressing uncertainty about effectiveness, dosage schedules, duration of protection, and eligibility criteria. Third, despite knowledge gaps, participants expressed anticipated improvements in HIV prevention through injectable PrEP, viewing it as a promising innovation that could address adherence challenges and offer greater client choice. Fourth, this optimism was contingent on demonstrated success, with many healthcare workers adopting a cautious stance pending evidence of real-world effectiveness in their campus settings.

These findings reveal that PHC workers occupy a position of informed hesitancy; they recognize injectable PrEP's potential benefits while expressing legitimate concerns about their preparedness to implement it safely and effectively. This state reflects neither uninformed resistance nor uncritical enthusiasm but rather professional responsibility in the face of inadequate preparation.

Beyond awareness and knowledge levels, PHC workers expressed diverse perceptions about injectable PrEP that warrant careful interpretation. A complex interplay of professional experiences, workplace contexts, and limited technological understanding shapes healthcare professionals' perceptions of emerging HIV prevention technologies. Multiple studies substantiate this claim, underscoring the critical role of knowledge and perception in shaping attitudes toward HIV prevention technologies [27–29]. For instance, Wheelock, *et al* [27] found that among 91 participants across seven countries, only 50 were aware of PrEP, with notable variations in support. Building on this, Rubincam *et al* [29] revealed that many participants equated a lack of 100% effectiveness with being unsafe, demonstrating how limited information impacts perception. Similarly, Kambutse, *et al* [28] specifically noted that over 50% of respondents had insufficient PrEP knowledge, though 86.5% of healthcare workers expressed interest. Taken together, these findings indicate that perceptions of new HIV prevention methods are nuanced, context-dependent, and heavily influenced by the quality and completeness of information available. Professional backgrounds significantly mediate understanding and acceptance of new HIV prevention methods.

The prevailing perception among healthcare workers was one of cautious optimism, a balanced stance that acknowledged both the potential benefits and the uncertainties surrounding the implementation of injectable PrEP. This tempered optimism is particularly noteworthy within the South African context, where injectable PrEP has not yet been integrated into public health programmes. In this environment, healthcare workers have formed complex, tempered perspectives on injectable PrEP shaped by fragmented information channels and evolving evidence. Empirical evidence supports this interpretation. *Asabor, et al* [30] found that healthcare workers emphasized structural barriers while simultaneously expressing cautious optimism about PrEP's potential to reduce HIV transmission. Similarly, Pleaner, *et al* [31] observed that many providers expressed concerns about possible unintended consequences, with many worried that PrEP might encourage risky sexual behaviour. This explains why their perceptions varied considerably and why many expressed reservations alongside their recognition of potential benefits.

In countries where new health technologies are introduced without prior local implementation, healthcare workers often form perceptions based on incomplete information, leading to knowledge gaps and uncertainties [6]. South Africa's position as a country preparing for but not yet implementing injectable PrEP means that PHC workers are uniquely positioned, aware of the innovation's existence and potential, yet lacking the experiential knowledge that comes from actual service delivery. This context of anticipatory readiness, rather than active implementation, fundamentally shapes their perceptions and explains the variations in knowledge and the cautious tone that characterized many responses.

South African PHC workers are in a state of anticipatory readiness for injectable PrEP, characterized by awareness of the innovation's potential but limited experiential knowledge. The evidence strongly supports this nuanced position. Femi-Lawal, Olawuyi [32] found high PrEP awareness (85%) among healthcare workers, but poor knowledge (18%) and only moderate positive attitudes (46%). Mbatha, *et al* [33] confirms the context of preparation, noting the need to map PHC workers' perspectives on injectable PrEP systematically. Pleaner, *et al* [31] further illustrate this cautious stance, with providers expressing concerns about adding a new service and worrying about potential unintended consequences. The data suggest a complex landscape of anticipation, where healthcare workers are cognizant of PrEP's potential but remain hesitant due to limited implementation experience.

Despite the uncertainties and knowledge gaps, healthcare workers expressed notably optimistic perceptions about injectable PrEP's potential to enhance HIV prevention. These positive perceptions centred on the convenience and long-acting nature of the injectable formulation, which participants viewed as key advantages that could improve client uptake and adherence compared to daily oral PrEP. Healthcare workers recognize significant challenges in daily oral PrEP adherence among students, primarily stemming from complex social and personal barriers. Multiple studies across African contexts reveal persistent obstacles. Research by Kayesu, *et al* [34] documented how adolescent girls and young women in Uganda struggled with medication concealment and privacy concerns. Similarly, Pintye, O'Malley [35] found that Kenyan adolescent girls cited forgetfulness, stigma, and disrupted routines as primary adherence challenges. Beesham, *et al* [36] further demonstrated that pregnant and postpartum women in Cape Town experienced structural barriers, including pill burden, side effects, and logistical difficulties in maintaining daily regimens. Collectively, these findings underscore that adherence challenges extend beyond individual motivation to encompass complex sociostructurally determinants.

One of the most prominent challenges is the difficulty of maintaining consistent daily pill-taking routines. Students struggle with consistent medication adherence, often finding daily regimens difficult to maintain [35]. Another critical barrier involves stigma. PrEP use is frequently associated with negative social perceptions, including assumptions of promiscuity or HIV status [37]. Such perceptions not only deter individuals from initiating PrEP but also contribute to discontinuation among current users. In addition to stigma, privacy concerns further complicate adherence efforts. Adolescents and young women face significant challenges in concealing medication use, particularly from family and partners [37]. Collectively, evidence from multiple qualitative studies across diverse populations underscores the intricate social dynamics that influence PrEP uptake and adherence. These findings reveal that, beyond biomedical efficacy, the success of PrEP interventions depends on addressing social, cultural, and relational factors that shape individuals' daily experiences and health behaviours.

Participants perceived injectable PrEP as an innovation that could particularly benefit individuals who struggle with the demands of daily oral medication. The perception that a bi-monthly injection could eliminate the burden of daily adherence decisions reflects healthcare workers' understanding of the behavioural and contextual barriers their clients face. This view aligns with implementation science perspectives on innovation adoption, where perceived relative advantage, the degree to which an innovation is seen as better than existing options, is a critical determinant of acceptance [24]. Healthcare workers' optimism suggests they perceive injectable PrEP as offering meaningful advantages over oral PrEP for specific populations and contexts. Multiple HIV prevention methods provide critical autonomy and choice for users, mirroring established practices in contraceptive services. Studies consistently demonstrate that offering diverse PrEP options empowers individuals to select prevention methods aligned with their personal circumstances [38]. For instance, research among female sex workers revealed a strong preference for injectable PrEP due to reduced stigma and easier adherence [39]. Similarly, a study of contraception practices found that women's PrEP preferences often directly aligned with their existing birth control methods, underscoring the importance of personalized prevention choices [38]. The emerging consensus is that diversifying prevention methods is essential for meeting varied client needs, with multiple studies highlighting that method choice itself can be a powerful form of personal empowerment in HIV prevention.

This perception reflects a patient-centred care philosophy that emphasizes respect for client autonomy and the importance of tailoring services to individual preferences [40]. The ability to choose between formulations was seen as potentially increasing overall PrEP uptake by attracting individuals who might reject oral PrEP but find injectable PrEP acceptable, and vice versa. Healthcare workers also perceived injectable PrEP as a welcome addition to the HIV prevention toolkit, representing an expansion of options that could strengthen combination prevention approaches. This perception demonstrates an understanding that no single prevention method will be universally acceptable or effective for all individuals, and that diverse options are necessary to maximize reach and impact across heterogeneous populations [41]. The view that injectable PrEP could complement rather than replace oral PrEP suggests that healthcare workers conceptualize HIV prevention as requiring multiple strategies tailored to different needs, contexts, and preferences.

Healthcare workers consistently recognize that medication adherence is a complex, multilevel challenge that extends far beyond individual motivation. Robust evidence supports this perspective across diverse contexts. [42] identified structural barriers, including transportation difficulties, healthcare system navigation challenges, and economic constraints, as fundamental impediments to PrEP uptake among mid-South communities. Antonini, *et al* [43] comprehensively documented that 100% of reviewed studies identified structural barriers, including suboptimal logistics for pill-taking, side effects, and anticipated stigma. Taylor, *et al* [44] demonstrated through within-participant analyses that adherence fluctuates based on psychological states, social circumstances, and daily life disruptions, confirming the dynamic, multilevel nature of adherence challenges. These converging findings demonstrate that PrEP adherence is influenced by interconnected social, structural, and contextual factors rather than individual will-power alone.

However, it is crucial to note that these optimistic perceptions were formed in the absence of direct implementation experience. Healthcare workers had not administered injectable PrEP, observed its use among their clients, or witnessed its outcomes in their clinical settings. Their positive perceptions remained largely theoretical, based on anticipated rather than observed benefits and informed by their interpretation of international research findings and their extrapolation from experiences with other injectable medications such as contraceptives. This distinction between anticipatory optimism and evidence-based confidence grounded in local expertise is significant for implementation planning.

The anticipatory nature of these perceptions suggests that healthcare workers' optimism could either be reinforced or tempered once implementation begins, depending on actual experiences with efficacy, acceptability, side effects, operational feasibility, and client satisfaction. If early implementations produce positive observable outcomes, successful adherence, client satisfaction, HIV prevention, and manageable side effects, healthcare workers' optimism will likely strengthen and spread through professional networks, accelerating broader adoption. Conversely, if early implementations encounter significant challenges, frequent side effects, client dissatisfaction, operational difficulties, or adherence issues, initial optimism could erode, potentially leading to resistance to continued rollout.

In countries where new health technologies are introduced without prior local implementation, healthcare workers often form perceptions based on incomplete information [6]. This explains the observed variations in knowledge among PHC workers in this study. These variations align with previous research on healthcare worker preparedness for new HIV prevention technologies, which has consistently demonstrated that knowledge gaps exist even among professionals working directly in HIV prevention services [38,45].

The critical factor explaining these knowledge variations is the absence of injectable PrEP implementation in South Africa at the time of this study. Without formal training programmes, clinical guidelines, or hands-on experience, PHC workers acquired knowledge through informal channels, including online news articles, international research reports, and conversations with colleagues who had heard about injectable PrEP. This informal knowledge acquisition process inevitably produces uneven and incomplete understanding. South Africa is in the preparatory phase of the injectable PrEP rollout, with regulatory approvals obtained but widespread implementation not yet underway. This preparatory context creates a knowledge environment in which healthcare workers are aware that change is coming but lack the structured information and training to feel confident and competent.

This is consistent with Mgodi, *et al* [6], who found that healthcare providers in sub-Saharan Africa often lack comprehensive knowledge about long-acting injectable PrEP despite its proven efficacy in clinical trials. The gap between research evidence and practice knowledge represents a critical barrier to the dissemination of new health innovations in the health sector. The variations observed among nurses, doctors, and HIV counsellors reflect differential access to information, different professional training pathways, and varying levels of engagement with emerging HIV prevention literature, all operating in a context where no formal, structured knowledge dissemination about injectable PrEP has yet occurred through official health system channels.

The limited awareness among nurses is particularly significant given their central role in primary healthcare service delivery in South Africa. Nurses are often the first point of contact for clients seeking HIV prevention services and are responsible for counselling, education, and medication administration. This gap may hinder the successful rollout of injectable PrEP, as nurses' confidence and competence are essential for effective implementation [17,46,47]. Similarly, the surface-level knowledge among HIV counsellors raises concerns, as they play a crucial role in helping clients make informed decisions about HIV prevention options. Their ability to accurately explain benefits, risks, administration schedules, and eligibility criteria is fundamental to client uptake and adherence. Therefore, addressing these variations through structured national training initiatives will be essential to ensure consistent understanding and readiness across professional groups once injectable PrEP becomes available for use in South Africa.

Beyond basic awareness, the limited understanding of the clinical and operational dimensions of injectable PrEP, particularly regarding its effectiveness, duration of protection, dosage schedules, and eligibility criteria, mirrors the findings of Ntimani, *et al* [48], who observed that healthcare providers in resource-limited settings often face challenges in keeping up with rapidly evolving HIV prevention guidelines and emerging biomedical technologies.

The confusion about dosage schedules and the duration of protection reflects a broader challenge in translating clinical trial evidence into practical knowledge for healthcare delivery. While cabotegravir for PrEP has been extensively studied in clinical trials such as HPTN 083 and HPTN 084, which demonstrated its superior efficacy compared to daily oral PrEP [7,8], this evidence might not yet have been effectively disseminated to frontline healthcare workers in South African campus health settings. The gap between research evidence and practice knowledge represents a critical barrier to the dissemination of new health innovations in the health sector. Since PHC workers are not yet implementing injectables, their uncertainty about clinical details is understandable and reflects the absence of formal training rather than any deficiency in their professional competence.

The uncertainty surrounding eligibility criteria suggests that PHC workers have not yet begun implementing injectable PrEP. However, this also highlights key areas that should be prioritized in preparatory training before rollout. Injectable PrEP has specific inclusion and exclusion criteria related to HIV status, kidney function, body weight, and concurrent medications [7,49]. Healthcare workers' inability to clearly articulate who qualifies for injectable PrEP at this preparatory stage is not concerning in itself; rather, it identifies an essential training need. This finding underscores the need for clear, accessible clinical guidelines that are tailored to the South African context and the specific needs of campus health settings, delivered through structured training before implementation begins.

## Conclusions

PHC workers recognize injectable PrEP's potential to enhance HIV prevention among university students, but require comprehensive, cadre-specific training and adequate resources before implementation. Policymakers should mandate site readiness assessments and allocate dedicated funding for comprehensive training programs before authorizing implementation.

## Supporting information

**S1 Appendix. Interview guide.** Semi-structured interview guide used to collect qualitative data from participants, including all interview questions and probes.
(PDF)

**S1 Checklist. COREQ checklist.** Completed 32-item Consolidated Criteria for Reporting Qualitative Research (COREQ) checklist (Tong et al., 2007), documenting methodological transparency across three domains: research team and reflexivity, study design, and data analysis and reporting.
(PDF)

## Acknowledgments

The authors gratefully acknowledge the primary healthcare workers who generously shared their time, experiences, and insights for this study. We thank the university administration and campus clinic leadership for facilitating this research. The authors acknowledge the use of Elicit (AI) to find articles.

## Author contributions

**Conceptualization:** Nomvuselelo Nomzamo Mbatha.

**Data curation:** Nomvuselelo Nomzamo Mbatha.

**Formal analysis:** Nomvuselelo Nomzamo Mbatha, Nomakhosi Mpofana, Dumile Gumede.

**Funding acquisition:** Nomvuselelo Nomzamo Mbatha.

**Investigation:** Nomvuselelo Nomzamo Mbatha.

**Methodology:** Nomvuselelo Nomzamo Mbatha, Nomakhosi Mpofana, Dumile Gumede.

**Supervision:** Nomakhosi Mpofana, Dumile Gumede.

**Validation:** Nomakhosi Mpofana, Dumile Gumede.

**Visualization:** Nomvuselelo Nomzamo Mbatha.

**Writing – original draft:** Nomvuselelo Nomzamo Mbatha.

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
