## [Decision Letter · Decision Letter 0]

2 Feb 2026

PGPH-D-25-03882

Campus Healthcare Workers' Knowledge and Views on Injectable Pre-Exposure Prophylaxis Prior to Rollout in South Africa: A Qualitative Study

Dear Dr. Nomvuselelo Nomzamo Mbatha

Thank you for submitting your manuscript to PLOS Global Public Health. After careful consideration, we feel that it has merit but does not fully meet PLOS Global Public Health’s publication criteria as it currently stands. Therefore, we invite you to submit a revised version of the manuscript that addresses the points raised during the review process.

We look forward to receiving your revised manuscript.

Kind regards,

Henry Zakumumpa, PhD

Academic Editor

Journal Requirements:

Additional Editor Comments (if provided):

We are delighted to share comments from our two reviewers. There is consensus from both reviewers around the need to strengthen the discussion section among the areas that need improvement.

Please endeavor to make the recommended corrections as much as possible so we can move swiftly to a decision

Reviewers' comments:

Reviewer's Responses to Questions

**Comments to the Author**

1. Does this manuscript meet PLOS Global Public Health’s publication criteria? Is the manuscript technically sound, and do the data support the conclusions? The manuscript must describe methodologically and ethically rigorous research with conclusions that are appropriately drawn based on the data presented.

Reviewer #1: Partly

Reviewer #2: Yes

2. Has the statistical analysis been performed appropriately and rigorously?

Reviewer #1: Yes

Reviewer #2: Yes

3. Have the authors made all data underlying the findings in their manuscript fully available (please refer to the Data Availability Statement at the start of the manuscript PDF file)?

Reviewer #1: Yes

Reviewer #2: Yes

4. Is the manuscript presented in an intelligible fashion and written in standard English?

Reviewer #1: Yes

Reviewer #2: Yes

5. Review Comments to the Author

Reviewer #1: This is a well written, clear article about HCW attitudes and knowledge around injectable PrEP, which is particularly pertinent for South Africa in view of the upcoming lenacapavir national roll out. I have a few comments that I'd like the authors to consider:

Background:

Lines 52-58. Cabotegravir is only one example of long acting injectable PrEP and is not available to the SA public beyond clinical trials, while Lenacapavir roll out is planned in South Africa for early 2026. The paragraph makes it sound like Cabotegravir will be rolled out and therefore I suggest discussing the efficacy of Lenacapavir from the Purpose 1 and 2 studies or speak about the efficacy of both products.

Methods:

Excellent description of qualitative methods – particularly trustworthiness section. Were participants reimbursed?

Results and discussion:

The results and discussion are well written, however the discussion in particular is quite lengthy and at times, it is only at surface level – more complexity is required. While authors have discussed the findings in relation to the broader literature, it would be good to have a bit more succinct summary and focus more on where the results fill in the gaps in literature.

However in saying this, while this is listed (not overtly) as a limitation, it is important to note that from what is reported in the results, none of the participants had experience in providing injectable PrEP or we assume, as it’s not publicly available, have also not worked in a facility that provides injectable PrEP. It therefore can be expected that they do not have any knowledge about it or have had any training on injectable PrEP provision – and this is not really adding to any gaps in knowledge in the HIV knowledge base.

As mentioned in the background, the gap reported is indeed correctly significant - campus health clinics face unique constraints, including fluctuating student demand driven by academic calendars, heightened privacy concerns within close-knit campus communities, and resource limitations specific to educational institutions. I would have liked to see more mention of these points in both the results and discussion. Is there more data around this population in particular? And the context? E.g. why Long acting PrEP would be better specifically in the student population? One of the worries around implementing long acting prep in tertiary institutions is the long student holidays and not returning for scheduled appointments. As long acting injectables have a long tail period and risk resistance to ARVs if seroconversion occurs – how will these risks be mitigated? Anticipated improvements are mentioned but what are the barriers or disadvantages? Many HCWs are already overwhelmed and can find new treatment regimes a burden, particularly when it is more medicalized than oral PrEP (and require more choice counselling and time when already overworked). It would be great to see more unpacking of the topic.

Reviewer #2: This is a well-presented piece of research, and very necessary to capture the perspectives of healthcare workers that serve this specific population. The paper would benefit from additional supporting evidence of the findings, and a tighter discussion that focuses on how the findings can inform implementation of long-acting injectable PrEP (and forthcoming HIV prevention technologies). Please see below for specific comments.

• P.2 line 28-29: “improvements in HIV prevention coverage”

• P.2 line 31: “dependent on the demonstrated success of injectable PrEP.” Unclear what that means.

• P.2 line 43: examples of national prevention programmes would be helpful

• P.2 line 43-44: are there certain contextual factors in KZN that contribute to the high HIV prevalence? Are young people disproportionately burdened by HIV in KZN similar to nationally?

• P.2 line 49: What does it mean for a young person to struggle with daily pill-taking due to lifestyle patterns?

• General intro: Really nice job setting up the public health problem and need for this work. While it is noted that South Africa has permitted a phased rollout of injectable, please specify to what extent injectable PrEP has been rolled out in KZN and at university clinics.

• P.5 line 101: different cadres of PHC – who does what related to PrEP?

• P.5 line 105: Please explain what is meant by “PHC workers with relevant knowledge and experience in HIV prevention service delivery.” Eligibility criteria part c indicates that PHC were eligible if they were directly involved in HIV prevention services. How was a PHC worker determined to have the “relevant knowledge and experience”?

• P.5 line 124: I do not think it’s relevant to specify the gender of the first author.

• P.6 line 128: What is meant by “all healthcare workers approached agreed to participate in the study” as it appears from earlier narrative that the researcher did not individually recruit participants – rather participants opted in to contacting researchers if they were interested in participating based on the flyer.

• P.6 line 132: was the guide piloted at non-study university health clinics?

• P.141 line 140-141: did both co-authors verify accuracy of transcripts?

• P. 6 line 142: please describe how Babchuk’s analysis process was utilized in this work.

• P.7 line 149: It would be helpful to define Lincoln and Guba’s 4 criteria before describing how these were achieved in this work.

• P.7 data analysis: More is needed to understand who conducted the data analysis – how many people were involved in the data analysis? How many people analyzed each transcript? Who developed the codebook? What was the role of the supervisor (Is the supervisor a co-author? If so, identify by initials.)

• P.7 line 157: recommend rephrasing to “comprehensive description of study context….”

• P.7 lines 158-161: Authenticity was ensured through fair representation of diverse perspectives, including minority views, and attention to power dynamics, ensuring marginalized voices were acknowledged.”

• P.7 line 172: separate out age and clinical experience as age is unrelated to career duration.

• P.7 participant characteristics: Please explain the role of these nurses, MDs, and HIV counsellors in HIV service delivery in their settings. Can nurses prescribe HIV prevention medication? Who can administer injections? Who conducts HIV tests?

• P.14 line 293: unclear what is meant by “acknowledging their doubts associated with the nature of the target population for injectable PrEP.” The corresponding quote seems to express concern about side effects and health outcomes.

• P.9 quote beginning line 199: seems better suited for theme 4.

• P. 11 line 226: It is noted that information is most often obtained through online news and peer networks. Please if possible describe further what “online news” means, and additionally, what other sources of information participants identified.

• P.11 line 226-228: this finding seems more related to theme 2.

• P.12 lines 257-259: please provide a quote to support the finding that limited understanding of eligibility guidelines created uncertainty and hesitation.

• P.13 line 276: It is not evident through the provided quotes how choice could empower users.

• P.13 theme 4: both quotes provided are from MDs. Are there supporting quotes from nurses or counsellors?

• Findings: In theme 1 findings, authors nicely describe how findings varied across cadres; however in findings for themes 2-4, this specificity is missing. While obviously these findings are not generalizable throughout each cadre, it would be helpful to identify if there were any differences/similarities across cadres.

• Findings: Generally, recommend providing more direct quotes – perhaps at least one per cadre – to support findings. The quotes are very rich!

• P.15 line 336-337: Recommend rephrasing to, “These findings reveal that PHC workers occupy a position of informed hesitancy, in that they recognize injectable PrEP’s potential benefits…”.

• P.16 first paragraph: Did PHC workers express diverse perceptions? The findings as presented seem to overwhelming suggest that providers were interested, optimistic, yet cautious about injectable PrEP. From the evidence presented (Wheelock, Rubincam, etc) – are these specific to injectable PrEP? Who are the participants in each study (healthcare workers?)?

• P.16 line 356-357: What is meant by “professional backgrounds significantly mediate understanding and acceptance of new HIV prevention methods”? Is it the cadre of PHC worker? The amount of experience in HIV prevention service delivery? Unclear how that statement connects to the findings presented here.

• P.17 line 368: Did this work have similar findings to Pleaner (unintended consequences, specifically encourage risky sexual behaviour)? Seems like this work’s findings are more in line with Asabor’s findings rather than Pleaner. Lines 370-772:

• P.17 line 384: Again, is the evidence in this paragraph specific to injectable PrEP?

• P.18 line 402: The evidence discussed here – is it specific to students? How is it relevant to the presented work?

• P.22 paragraph beginning at line 499 is a repeated paragraph (p. 17 line 374).

• Discussion: While evidence from other studies is brought into the discussion, it is not always clear how this evidence does or does not support or align with the current work. At times, the discussion seems more like a review of the literature, and less contextualizing the work within the current evidence base. Additionally, it is not clear if the evidence presented is relevant to the current work – for example, is it related to students, young people, healthcare workers at universities? More precise language is needed to connect the authors’ findings to established evidence. For example, p. 20, paragraph beginning with line 464: This paragraph is summary of evidence that identifies structural barriers to PrEP uptake. But how is this evidence relevant to the present work? Overall, the discussion includes a lot around client-level barriers, but less around healthcare workers perceptions – recommend streamlining to focus on healthcare workers perspectives (per the findings!). For example, the discussion around tempered optimism and informed hesitancy is helpful to understand as clinics, governments, health systems prepare to introduce a new technology – it would be useful to connect these findings to practical guidance for implementation. It would also be useful to discuss how these findings can inform future introduction of HIV prevention technologies (long-acting lenacapavir, for example).

• P.23 line 529: earlier in the paper, under theme 1, it is noted “Among the ten nurses interviewed, seven reported having some knowledge of injectable PrEP, whereas three indicated limited or no awareness of it.” This contradicts line 529.

• P.24 line 562: Previously, the authors note that injectable PrEP implementation has not yet begun – line 562 seems to say that this is due to uncertainty surrounding eligibility criteria. Is that factual? Or has it not begun for other reasons (not yet rolled out in South Africa)? This is not clear.

6. PLOS authors have the option to publish the peer review history of their article (what does this mean?). If published, this will include your full peer review and any attached files.

**Do you want your identity to be public for this peer review?** For information about this choice, including consent withdrawal, please see our Privacy Policy.

Reviewer #1: No

Reviewer #2: No

Figure Resubmissions:

---

## [Editor Report · Decision Letter 1]

21 Apr 2026

Campus Healthcare Workers' Knowledge and Views on Injectable Pre-Exposure Prophylaxis Prior to Rollout in South Africa: A Qualitative Study

PGPH-D-25-03882R1

Dear Nomvuselelo Nomzamo Mbatha,

We are pleased to inform you that your manuscript 'Campus Healthcare Workers' Knowledge and Views on Injectable Pre-Exposure Prophylaxis Prior to Rollout in South Africa: A Qualitative Study' has been provisionally accepted for publication in PLOS Global Public Health.

Best regards,

Henry Zakumumpa, PhD

Academic Editor